# Biological Pretreatment Effects on the Bioconversion of Brewer Spent Grain with *Hermetia illucens* Larvae

**Martha Sumba-Zhongor** [1,*], **Eduardo Álava** [1], **Luis Galarza** [2], **Johana Ortiz-Ulloa** [3], **Eduardo J. Chica** [4], **Omar Ruiz-Barzola** [1,5], **Julia Nieto-Wigby** [1], **Maria Isabel Jiménez-Feijoo** [1] and **Malena Torres-Ulloa** [1]

1   Faculty of Life Sciences, Escuela Superior Politécnica del Litoral, ESPOL, Campus Gustavo Galindo Km. 30.5 Vía Perimetral, Guayaquil 090112, Ecuador
2   Escuela Superior Politécnica del Litoral, ESPOL, Biotechnology Research Center of Ecuador, CIBE, Campus Gustavo Galindo Km. 30.5 Vía Perimetral, Guayaquil 090112, Ecuador
3   Department of Biosciences, Faculty of Chemical Sciences, University of Cuenca, Av. 12 de Abril s/n Cdla. Universitaria, P.O. Box 01.01.168, Cuenca 010203, Ecuador
4   Faculty of Agricultural Sciences, Universidad de Cuenca, Cuenca 010203, Ecuador
5   Department of Statistics, Faculty of Medicine, Universidad de Salamanca, 37007 Salamanca, Spain
*   Correspondence: mbsumba@espol.edu.ec

**Abstract:** *Hermetia illucens* is an important species for waste management and the circular economy. The aim of this study was to analyze the effects of *Trichoderma reesei* C2A and *Pleurotus* sp. as pretreatments of brewer spent grain (BSG). BSG was inoculated with fungal solution or distilled water (control). After seven days, this was used for *H. illucens* larvae cultivation. At the end of bioconversion process, parameters of substrate reduction and *H. illucens* larval development were evaluated. Chemical properties of BSG, frass and larvae were also analyzed. With *T. reesei* C2A pretreatment, highest substrate reduction ($46.3 \pm 0.9\%$) was achieved, but larval growth rate was lower ($1.0 \pm 0.1$ mg/d) than that of control ($2.8 \pm 0.2$ mg/d). Larvae of *Pleurotus* sp. pretreatment had limited development, reflected in their negative growth rate ($-0.6 \pm 0.2$ mg/d). In conclusion, cultivation of *H. illucens* larvae (six day old) on BSG pretreated with *Pleurotus* sp. is not recommended. On the other hand, *T. reesei* C2A pretreatment enhance BSG reduction, and its potential use for lignocellulosic waste management should be more explored.

**Keywords:** Agroindustry; black soldier fly; insect protein; microbes; organic waste management

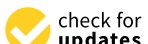



## 1. Introduction

Brewer's spent grain (BSG) is the main by-product of beer-brewing industry, it represents about 85% of the solid by-products [1,2]. Around 20 kg of wet BSG are obtained per hectoliter of beer produced [3]. According to this, the world annual production of BSG is 40 million tons [4], with almost 120,000 tons generated by Ecuador [5]. BSG is a lignocellulosic material with a diverse composition, containing fiber (30–70%), protein (20–30%), lipids ($\approx$10), and minerals [2,6,7].

Traditionally, BSG has been used as cattle feed, however, due to their high moisture (77–81%) and sugar content is susceptible to microbial spoilage and so often discarded improperly with negative impacts on the environment [6–8]. During the last ten years, some studies have been conducted with the aim to develop alternative strategies for its recycling and revalorization. The most basic strategy could be composting, although due to its high moisture and physicochemical properties, it must be mixed with other organic materials to achieve optimal conditions for its biodegradation [9]. Moreover, BSG has demonstrated to be a good substrate for lignocellulolytic enzymes production [2], and cultivation of edible mushrooms [10,11] and insects such as *Hermetia illucens* larva [12,13].

*H. illucens* has captured great attention throughout the world due to its bio converting capacity of organic residues into value added bioproducts. *H. illucens* larvae can grow on

several organic residues coming from animal, plants, and human sources, attaining high proportions of protein (37–63% dry matter; DM) and fat (37–63% DM) useful for animal feeding [14]. Additionally, *H. illucens* larva manure (frass), a byproduct of the bioconversion process, is a potential source of macro and micronutrients [15], and so currently marketed as a soil amendment.

Though *H. illucens* larva cultivation, BSG can be reduced by 45% DM basis, but non-lignocellulosic substrates, such as food remains, could be reduced up to 82% DM basis [13]. In some studies, it has been seen that pretreatments are a key part in biotechnological applications for breaking down lignocellulose into fermentable simple sugars [16].

Recent studies related to *H. illucens* bioconversion have shown that a biological pre-treatment alone or in combination with a mechanical pretreatment improves the bio-conversion of different organic residues, such as maize straw [17], banana peels [18], chicken manure [19], and coconut pulp [20]. In these studies, biological pretreatment with *Aspergillus niger* [17], *Rhizopus oligosporus* [18], *B. subtilis* [19], and yeast [20] improved the substrate reduction and *H. illucens* larval development. Additionally, fungi, such as *Trichoderma reesei*, and *Pleurotus* sp., are well recognized for their capacity to produce extracellular enzymes which favor lignocellulose biomass decomposition [21,22].

Due to the capacity of *T. reesei* and *Pleurotus* sp. to breakdown fibrose organic residues, the hypothesis of this study is based on the fact that pretreatment of the substrate improves the bioconversion of BSG. The objective of this study is to analyze the effects of their use on the reduction of BSG and *H. illucens* larval development.

## 2. Results

### 2.1. Experiment One

To limit proliferation of opportunistic microbes during pretreatment, BSG was steril-ized. Then, the effects of fungi on sterilized and non-sterilized BSG were evaluated both in the absence and presence of *H. illucens* larvae.

Higher substrate reduction (SR) percentages were obtained in the presence of *H. illucens* larvae no matter if BSG was sterilized or not ($\approx$50%) ($p < 0.0001$). But, when BSG was sterilized and larvae were absent (Control 1 and 2), lower SR were obtained (7.8 and 9.3%) ($p < 0.0001$) (Figure 1).

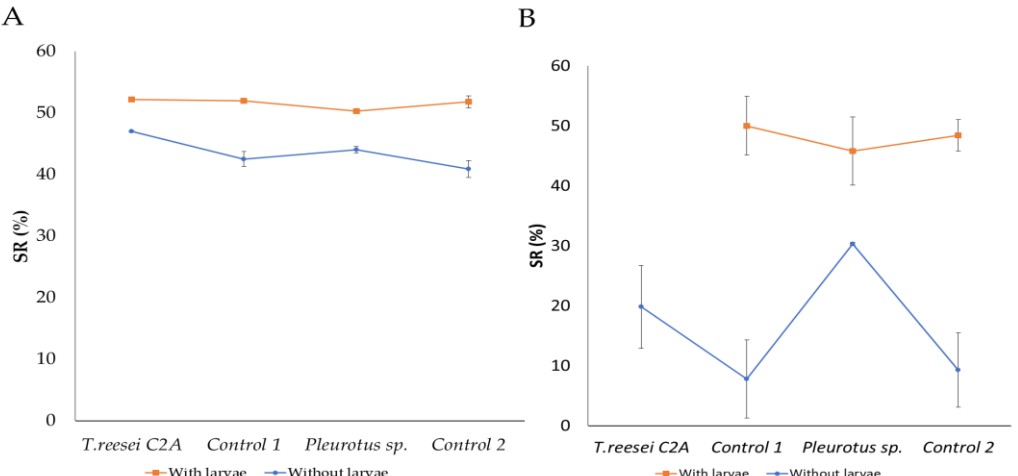

**Figure 1.** Substrate reduction (SR) of (**A**) non-sterilized and (**B**) sterilized BSG, after pretreatments with *T. reesei* C2A and *Pleurotus* sp., and *H. illucens* larva cultivation. Symbols represent the mean (*n* = 3) and error bars the standard deviation of the mean (Tukey test, $p < 0.0001$). Control 1 and 2: BSG without inoculation of fungi.

### 2.2. Experiment Two

Larvae from *T. reesei* C2A treatment grew more slowly than the control ($p < 0.0001$) (1.0 $\pm$ 0.1 mg/d vs. 2.8 $\pm$ 0.2 mg/d, respectively) (Figure 2A), but the survival rate was

similar on both cases ($p > 0.05$). ($98.3 \pm 2.9\%$ and $99.5 \pm 1\%$) (Figure 2B). Larvae fed on BSG that was pretreated with *Pleurotus* sp. did not develop normally. This treatment was suspended at tenth day with none prepupa, although 70% of the larvae survived (Figure 2B), they showed a negative growth rate (GR, $-0.6 \pm 0.2$ mg/d) (Figure 2A). Regarding crude protein content (CP) of the larvae, there were no statistical differences between *T. reesei* C2A and the control ($42.6 \pm 1\%$ and $43.1 \pm 0.6\%$) ($p = 0.34$) (Table 1). CP of larva from *Pleurotus* sp. treatment was not analyzed since not enough larval biomass was obtained.

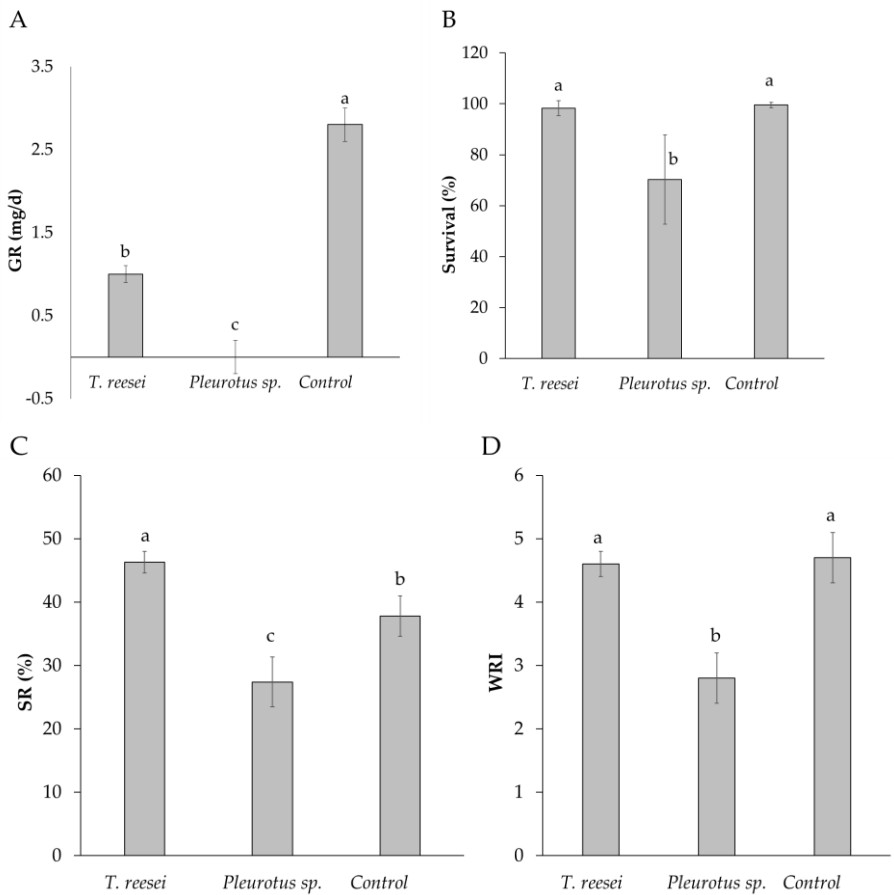

**Figure 2.** (**A**) Larval growth rate (GR), (**B**) Survival, (**C**) Substrate reduction and (**D**) Waste reduction index after cultivation of *H. illucens* larvae on BSG pretreated with *T. reesei* C2A and *Pleurotus* sp. Means with different letters are significantly different according to Tukey or Dunn test, $p < 0.05$ ($n = 4$, mean $\pm$ SD). Control: without inoculation of fungi.

**Table 1.** Chemical properties of larvae and frass from *H. illucens* ($n = 4$, mean $\pm$ SD).

| | Larva | Frass | |
| --- | --- | --- | --- |
| | CP (%) | C/N Ratio | pH |
| *T. reesei* C2A | $42.6 \pm 1$ | $14.7 \pm 1.0$ | $5.4 \pm 0.1$ |
| *Pleurotus sp.* | N.d. | N.d. | N.d. |
| Control | $43.1 \pm 0.6$ | $15.9 \pm 1.9$ | $5.3 \pm 0.1$ |

Control: without inoculation of Fungi. N.d.: Not determined.

Higher substrate reduction (SR) was achieved with *T. reesei* C2A when compared to the control C1 ($46.3 \pm 0.9\%$ vs. $37.8 \pm 1.6\%$) ($p = 0.003$) (Figure 2C). When waste reduction index (WRI) was analyzed, there were no statistical differences between *T. reesei* and Control 1 ($4.7 \pm 0.4$ and $4.6 \pm 0.2$ respectively) ($p < 0.0001$) (Figure 2D). Lowest SR and WRI were obtained with *Pleurotus* sp. ($27.4 \pm 3.9\%$ and $2.8 \pm 0.4$, respectively) ($p < 0.0001$) (Figure 2C,D).

There were not statistical differences in carbon nitrogen (C/N) ratio nor pH of the frass between treatments and control ($p > 0.05$), but the C/N ratio and pH of frass were higher than those of the substrate (Table 1).

## 3. Discussion

This study evaluated for the first time, the effects of *T. reesei* C2A and *Pleurotus* sp. pretreatments on substrate reduction and larval development of *H. illucens* cultivated on the BSG pretreated. Experiment number one showed substrate reduction of BSG was higher due to the presence and activity of *H. illucens* larvae whether BSG was sterilized or not. In the absence of *H. illucens* larvae, SR of non-sterilized BSG was higher than that of sterilized BSG, probably due to the proliferation of opportunistic microbes that showed up during the cultivation period. Because of non-inoculated fungi were observed on the controls, sterilized substrate was employed for a second experiment.

The pretreatment with *T. reesei* C2A for seven days prior to the culture of *H. illucens* led to a 22% greater SR when compared to the control. This interesting finding results very important for managing BSG residue of the beer producing process, which is commonly discarded to the environment with subsequent polluting effects. Although higher amount of BSG was reduced, it was not necessarily due to the addition of BSF larvae, as they showed low evidenced in their lower growth rate.

Taking into consideration high CP of BSG (26.9%), pretreatments with mentioned fungi seems not improve larval development. It seems that a pretreatment application of less nutritive substrates could be more efficient. This observation agrees with results obtained by Isibika et al. who found that banana peels (68.6% of fiber, 5.5% of protein), had 50% less crude fiber content after seven days of fermentation with a *T. reesei* strain [18]. In the mentioned study, the pretreatment aided the decomposition of the substrate as well as the increase of *H. illucens* larval weight under continuous feeding regimen. However, in our study, substrate was supplied at once, so water or nutrients could be loss during the process.

The treatment with *T. reesei* C2A did not influence the protein content of *H. illucens* larvae. The amount of protein found was similar to values reported by Nyakeri et al. (43%) but inferior to those reported by Liu et al. (49.9–54.1%) for the same type of substrate [13,23]. A possible explanation for the variation in content of protein of the larval biomass among studies is the variability of the nutritional profile of the BSG as a subproduct.

As for *Pleurotus* sp. treatment, larvae presented abnormal behavior, migrating to the perimeter of the tray. None reached the prepupae stage, therefore the treatment was suspended at day 10 of the experiment, by this time the substrate looked dry and compact. On a previous study undertaken by the same authors, older *H. illucens* larvae were fed BSG pretreated for three days with *Pleurotus* sp., completing their larval phase with a survival rate of 100%. A possible explanation for this different performance is that the larvae consumed the substrate before it was completely colonized by the fungi mycelium. This study corresponds to the first report of co-culture of *Pleurotus* sp. and *Hermetia illucens* on BSG.

Based on the results obtained in the second experiment, it can be clearly seen that the rise of C/N ratio and pH was mainly due to *H. illucens* larval activity rather than the inoculated fungal activity. This observation agrees with results from the studies of Meneguz et al. and Rehman at al., where *H. illucens* larva was able to increase pH from 4 to 9 [24] and from 5.7 to 8.6 [25]. Regarding C/N ratio, contrast results were obtained by other authors, which may be due to substrate properties, especially nitrogen content [25,26].

Finally, it is important to mention that the protocols to determine harvest age, post-harvest larval and the factor for protein quantification influence the results. Method standardization is key in *H. illucens* bioconversion studies to compare results among studies, using, when possible, procedures suggested by Bosch et al. [27].

## 4. Materials and Methods

### 4.1. Collection and Rearing of Hermetia illucens

Wild *H. illucens* larvae were collected in an urban sector of the city of Guayaquil, Guayas province, Ecuador. The collected larvae were used to establish a colony in a nursery garden at the Life Sciences Faculty of the Escuela Superior Politécnica del Litoral (ESPOL) (108 m.a.s.l.). Resulting flies were housed in a metallic structure ($0.75 \times 0.75 \times 1.5$ m) covered with mosquito net and kept under ambient conditions (temperature 21–31 °C). A tray with fermented fruits (mainly mango) served as an attractant substrate, and cardboard pieces ($5 \times 3$ cm) located above the feeding tray aided oviposition. Eggs were daily collected and transferred to plastic containers until hatching (3rd to 4th d after collection). Neonate larvae were fed ad libitum with chicken feed meal (60% humidity) for five days according to Bosch et al. [27]. On the sixth day, larvae were separated from the feed using a fine paintbrush. Larvae assigned to the experiments corresponded to the first and second generation produced under laboratory conditions. Unused larvae were used for maintenance of the colony.

### 4.2. Fungi Culture

The fungus, *T. reesei* C2A and *Pleurotus* sp. were provided by Biotechnology Research Center of Ecuador (CIBE). *T. reesei* C2A was cultured in Potato Dextrose Agar (PDA media), and *Pleurotus sp.* was cultured in PDA media supplemented with 15 g of malt extract and 5 g of mycological peptone per 1,000 mL of distilled water. After incubation for seven days at 28 °C, spores were rinsed with a sterile saline solution (NaCl 0.9%) and diluted to a final concentration of $1 \times 10^8$ CFU/mL [18]. Spores were counted using a Neubauer chamber, and resulting suspensions were stored at 4 °C for later use.

### 4.3. Study One

Glass bottles were filled with hydrated BSG (9 g of BSG: 12 mL of water) and covered with aluminum foil. Then, half of the bottles were subjected to sterilization by autoclaving at 121 °C for 15 min. Some of the bottles with sterilized and non-sterilized BSG were inoculated with *T. reesei* C2A and *Pleurotus* sp. Covered with cotton plugs and incubated for three days at 28 °C. After the incubation period, ten *H. illucens* larvae were added to some bottles prior to incubation for 14 days at 28 °C. Then, larvae were harvested, and residues dehydrated for substrate reduction evaluation. All treatments had three replicates.

### 4.4. Study Two

#### 4.4.1. Substrate Preparation

BSG was mixed with water at a ratio of 70 mL: 30 g of BSG and autoclaved at 121 °C for 15 min. This material was stored at 4 °C until the next day.

#### 4.4.2. Experimental Design

This study was designed to determine the effects of *T. reesei* C2A and *Pleurotus* sp., pretreatments of BSG with the subsequent cultivation of *H. illucens* larvae. All treatments were run using four replicates.

Hence, 100 g of sterilized BSG (70% humidity) was placed in plastic trays ($\Phi$ 10 cm, h 6 cm), immediately one ml of spore solutions ($1 \times 10^8$ CFU/mL), or distilled water (control) was added to the trays. After seven days of fermentation, 100 six-day old larvae (average weight 2.7–8.7 mg) where added to trays (density of 1.27 larvae/cm$^2$). Each unit was covered with mosquito mesh netting (0.5 mm) and incubated at 28 °C. As soon as the first prepupa was identified, larvae were harvested using entomological tweezers. Larval weight was measured following the methodology described by Somroo et al. [28] with slight modifications. Collected larvae were kept unfed for 24 h to empty their intestinal tract, after which they were rinsed with tap water and dried with paper towels. Once clean, they were weighed and inactivated at 105 °C for 10 min. The dry weight of larvae and residues of the process were determined after drying the samples for two days at 60 °C.

### 4.4.3. Larval Development and Substrate Reduction Parameters

Larval growth rate (GR), survival, substrate reduction (SR), and waste reduction index (WRI) were calculated as in the work of Bosch et al. [27] and Diener et al. [29], using the following formulas:

$$\text{GR, mg/d} = \frac{\text{Average FLW} - \text{Average ILW}}{\text{Rearing time}} * 100 \tag{1}$$

$$\text{Survival, \%} = \frac{\text{Amount of live larvae at the end of process}}{\text{Amount of larvae sowed}} * 100 \tag{2}$$

$$\text{SR (\%)} = \frac{S - R}{S} * 100 \tag{3}$$

$$\text{WRI,} = \frac{S - R}{S * \text{rearing time}} * 100 \tag{4}$$

where: GR = larval growth rate, FLW = Final weight of larval biomass, ILW = Initial weight of larval biomass, SR = Substrate reduction, Rearing time = larval development time measured from the beginning of the experiment to harvest, S = total substrate weight administrated at the start of the experiment, and R = weight of residues collected at the end of experiment. All parameters were measured on dry basis.

### 4.4.4. Chemical Analysis

Dried samples of larvae, substrate, and residues were grinded with a coffee mill (Krups$^{®®}$ F203) for 20 s. Total nitrogen of all samples was determined by combustion using an elemental organic analyzer (Elementar$^{®®}$ Vario MACRO cube). To obtain the crude protein content (CP) of larvae, the resulting nitrogen value was multiplied by the protein conversion factor of 4.76 proposed by Janssen et al. [30]. Substrate and residues were additionally analyzed for total carbon with the same equipment (Elementar$^{®®}$ Vario MACRO cube). pH was measured with a potentiometer (Thermo Scientific $^{®®}$ ORION STAR A215).

The crude protein content, C/N ratio, and pH of BSG (dry matter basis) was $26.9 \pm 1$, $10.1 \pm 0.2$, and 4.2, respectively.

### *4.5. Statistical Analysis*

All data were examined with Shapiro–Wilk, Levene, and Durbin–Watson test, for normality, homoscedasticity, and variance independence assumption check compliment, respectively. A one-way analysis of variance (ANOVA) test was used to analyze the resulting data. Survival parameter did not comply with normality assumption, so Kruskal-Wallis nonparametric test was applied. GR and WRI did not fulfill with homogeneity of variance assumption, so Welch's correction was applied to the ANOVA test. When significant differences were found, the means were compared using Tukey test (parametric) or Dunn test (non-parametric) with a significance level of 0.05 ($p < 0.05$). Statistical analyses were performed using R 4.2.2 software.

## 5. Conclusions

The application of *T. reesei* C2A as pretreatment for seven days with the subsequent inoculation of *H. illucens* larvae lead to a major BSG reduction. The percentage of substrate decomposed was apparently not consumed by the *H. illucens* larvae affecting the larval growth. Nevertheless, the larvae were able to coexist with *T. reesei* C2A, reaching the harvest age in a short period, which was not the case with the *Pleurotus* sp. treatment, in which a limited development of the larvae was observed. Further studies are necessary regarding the application of *T. reesei* C2A and other fungal strains as a pretreatment of lignocellulosic substrates with low nutritional value (protein content). It is advisable to evaluate different concentrations of microbial complexes as well as fermentation times as to establish optimal

fermentation conditions according to the type of substrate studied. Cultivation of *H. illucens* larvae (six day old) on BSG pretreated with *Pleurotus* sp. is not recommended.

**Author Contributions:** Conceptualization, M.S.-Z., E.Á., E.J.C., J.O.-U. and L.G.; methodology, M.S.-Z., E.Á. and L.G.; formal analysis, M.S.-Z., E.Á. and O.R.-B.; investigation, M.S.-Z.; resources, L.G., M.I.J.-F. and M.T.-U.; writing—original draft preparation, M.S.-Z.; writing—review and editing, E.Á. and J.N.-W.; supervision, E.Á. All authors have read and agreed to the published version of the manuscript.

**Funding:** This research received funding from Escuela Superior Politécnica del Litoral (ESPOL).

**Data Availability Statement:** The data present in this study are available on request from the corresponding author.

**Acknowledgments:** The authors kindly acknowledge to CIBE for kindly provided the microorganisms used in the study. We are also grateful with Blanca Sumba for providing *Hermetia illucens* larvae and PRODAL S.A. for the BSG donation. This research was undertaken as part of a joint graduate program within the VLIR NETWORK from Ecuador.

**Conflicts of Interest:** The authors declare no conflict of interest.

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
