# Peer review of "Biological Pretreatment Effects on the Bioconversion of Brewer Spent Grain with Hermetia illucens Larvae"

_recycling, doi:10.3390/recycling8010003_

Round 1
Reviewer 1 Report
The biomass of larvae from brewer should be analyzed, such as the content of the amino acid and fatty acid。 Conversion process is needed also, such as temperture and so on.Author Response
- Does the introduction provide sufficient background and include all relevant references? can be improved
New relevant references were added to the introduction
- Are the methods adequately described? can be improved
Methods description was modified, part of a previous study was added.
- Are the results clearly presented? can be improved
Presentation of results was improved, there are also more figures.
- The biomass of larvae from brewer should be analyzed, such as the content of the amino acid and fatty acid
We mainly focus on the management of the substrate rather than larval biomass production. Although we include survival and larval growth rate.
- Conversion process is needed also, such as temperture and so on.
In line 238 we mention that process was developed at 28 Celsisus degree. Relative humidity was not measured at the incubator since we don't have a termohigrometer.
Reviewer 2 Report
Moderate English changes required
Add mored cited references relevant to the research
https://www.mdpi.com/1996-1073/15/10/3544
Author Response
- Does the introduction provide sufficient background and include all relevant references? can be improved
New relevant references were added to the introduction
- Are all the cited references relevant to the research? Must be improved
New relevant references were added to the introduction
- Are the results clearly presented? can be improved
Presentation of results was improved, there are also more figures.
- Are the conclusions supported by the results?
Yes, it was slightly modified.
- Add mored cited references relevant to the research https://www.mdpi.com/1996-1073/15/10/3544
This and other relevant references were added to the article.
Reviewer 3 Report
1. Please describe the novelty of this research.
2. This article only has three figures. Please add more results in this manuscript.
3. I think you should cite more reference.
Author Response
- Please describe the novelty of this research.
The novelty is previously described on the first paragraph
- This article only has three figures. Please add more results in this manuscript.
More figures were added
- I think you should cite more reference.
New relevant references were added to the introduction
Reviewer 4 Report
In general, the idea is good but the result are not so clear. I will suggest to repeat the experiment with the suggestion I will include in the conclusion. The idea was very good , the execution not so more. I’m sorry for that.
The idea is very original, but more work is needed for having scientific result of interest.
You could find more ideas in the attached document.

Round 2
Reviewer 3 Report
This manuscript is better than before. Please check all the details in the article again. For example, the Figure 4, Figure 5, and Table 1 need a period.
Author Response
Figures were grouped and details were revised.
We considered observations from revisor 4 about an error on methodology with B. subtillis and decided to remved that assay.
Reviewer 4 Report
You did a good job for improving, but I found a last issue. When you performed the trial you inoculated B.subtilis in BSG and immediately you add the larvae. So the larvae ate the bacterias before they could grow.
The B.subtilis need 37°C for 24 hour for developing ion a new media. You did not respect this time in the experimental design. You use the agar for multiplicate the bacteria at 37°C but you forgot to do the same in the diet before to provide to the larvae. or exclude this treatment or you should repeat it.

Author Response
Thanks for observations and suggestion about Bacillus subtilis. We agree and decided to removed bacterial assay.